# The carcinine transporter CarT is required in *Drosophila* photoreceptor neurons to sustain histamine recycling

Drew Stenesen, Andrew T Moehlman, Helmut Krämer*

Department of Neuroscience, University of Texas Southwestern Medical Center, Dallas, United States

**Abstract** Synaptic transmission from *Drosophila* photoreceptors to lamina neurons requires recycling of histamine neurotransmitter. Synaptic histamine is cleared by uptake into glia and conversion into carcinine, which functions as transport metabolite. How carcinine is transported from glia to photoreceptor neurons remains unclear. In a targeted RNAi screen for genes involved in this pathway, we identified *carT*, which encodes a member of the SLC22A transporter family. CarT expression in photoreceptors is necessary and sufficient for fly vision and behavior. Carcinine accumulates in the lamina of *carT* flies. Wild-type levels are restored by photoreceptor-specific expression of CarT, and endogenous tagging suggests CarT localizes to synaptic endings. Heterologous expression of CarT in S2 cells is sufficient for carcinine uptake, demonstrating the ability of CarT to utilize carcinine as a transport substrate. Together, our results demonstrate that CarT transports the histamine metabolite carcinine into photoreceptor neurons, thus contributing an essential step to the histamine–carcinine cycle.

*For correspondence: helmut.kramer@utsouthwestern.edu

Competing interests: The authors declare that no competing interests exist.

## Introduction

Histaminergic neurotransmission plays an important role in a variety of mammalian and invertebrate behavioral processes and contributes to the regulation of arousal, sleep and circadian rhythms (*Nall and Sehgal, 2014*; *Panula and Nuutinen, 2013*). Defects in histaminergic signaling are linked to multiple neurodegenerative diseases, depression and Tourette's syndrome (*Panula and Nuutinen, 2013*; *Shan et al., 2015*; *Castellan Baldan et al., 2014*). Synaptic clearance and recycling of monoamine neurotransmitters, including serotonin and dopamine, depend on transporters that promote uptake into the presynaptic terminals or the surrounding glia (*Torres and Amara, 2007*). For histamine, such transporter activities have been documented in mammalian and invertebrate glia (*Yoshikawa et al., 2013*; *Edwards and Meinertzhagen, 2010*). In the *Drosophila* visual system, histidine decarboxylase (Hdc) generates histamine in photoreceptor neurons (*Burg et al., 1993*). Light-induced depolarization of these neurons (*Wang and Montell, 2007*) promotes synaptic release of histamine, which opens histamine-gated chloride channels on postsynaptic L1 and L2 lamina neurons, triggering their hyperpolarization (*Pantazis et al., 2008*). These postsynaptic voltage changes can be followed with electroretinograms (ERGs) that reveal ON and OFF transient peaks coinciding with initiation and cessation of the light source (*Alawi and Pak, 1971*; *Heisenberg, 1971*). In addition to identifying mutants that interfere with presynaptic release (*Kim et al., 2012*), this phenotype has also contributed to the genetic dissection of histamine recycling (*Edwards and Meinertzhagen, 2010*). Synaptic histamine is taken up into epithelial glia that completely envelope photoreceptor synapses (*Meinertzhagen and O'Neil, 1991*). Within these glia, histamine is modified by the Ebony-catalyzed condensation with β-alanine (*Richardt et al., 2002*; *Borycz et al., 2002*; *Hartwig et al., 2014*; *Richardt et al., 2003*). The reaction product carcinine is transferred to photoreceptors, which

**eLife digest** Photoreceptors are light-sensitive neurons in the eyes of the fruit fly *Drosophila* that form connections with other neurons in the fly's brain. At these connections, which are called synapses, the photoreceptors continuously release a chemical called histamine.

Photoreceptors will release more or less histamine depending on changes in light intensity, but always tend to release more histamine than they can produce themselves from scratch. This means that the visual system in *Drosophila* relies on a pathway that recycles histamine. That is to say, glial cells (which support the activity of the neurons) remove the chemical from synapses and return it to the photoreceptor neurons in a slightly modified form called "carcinine". The photoreceptors then quickly convert the chemical back into histamine, ready to be released.

Stenesen et al. set out to identify the proteins that support this recycling pathway, and started by screening around 130 genes that encode transporter proteins for potential roles in histamine recycling. This screen identified a gene encoding a protein that was named CarT. This protein transports carcinine, the modified version of the histamine neurotransmitter.

Stenesen et al. show that the photoreceptor neurons make the CarT protein and need this protein to take up the carcinine released by the supporting glial cells. Without CarT, photoreceptor neurons cannot transmit visual information, and so mutant flies in which the gene for CarT is deleted are blind. Follow-up studies related to this work could involve identifying the transporters that move histamine and carcinine in and out of the glia cells, and exploring what other neurons and behaviors in fruit flies rely on CarT's activity.

recover histamine by Tan-catalyzed hydrolysis (*Borycz et al., 2002*; *True et al., 2005*; *Wagner et al., 2007*). This histamine–carcinine cycle (see below: *Figure 4—figure supplement 1*), reminiscent of the glutamate–glutamine cycle in the mammalian brain (*Bröer and Brookes, 2001*), is necessary to maintain visual neurotransmission (*Borycz et al., 2005*) and depends on the compartmentalization of glial Ebony and neuronal Tan (*Stuart et al., 2007*). Mutants for *ebony* or *tan* lack ON and OFF transients (*Heisenberg, 1971*; *Hotta and Benzer, 1969*; *Chaturvedi et al., 2014*), indicating that neither histamine synthesis in photoreceptor neurons by Hdc nor a putative direct re-uptake mechanism is sufficient to sustain neurotransmitter release at photoreceptor synapses (*Ziegler et al., 2013*). The identity of the transporters that facilitate glial uptake of histamine from the synaptic cleft and transport of its metabolite, carcinine, out of glia and into photoreceptor neurons remains a long-standing mystery, (*Edwards and Meinertzhagen, 2010*; *Stuart et al., 2007*; *Romero-Calderón et al., 2007*; *2008*). Here, we show that the previously uncharacterized *CG9317* gene, that we named *carT*, encodes a transporter responsible for uptake of carcinine into photoreceptor neurons. We demonstrate that CarT activity is a prerequisite for visual neurotransduction.

## Results and discussion

### *CG9317* knockdown in photoreceptor cells disrupts visual signal transduction

The *Drosophila* genome is estimated to contain 603 transmembrane transporters (*Ren et al., 2007*; *Featherstone, 2011*). To concentrate on those transporters with a higher probability of participating in histamine neurotransmitter recycling, we focused on families that contain individual members previously associated with neurotransmitter transport. This includes known serotonin, dopamine, gamma-aminobutyric acid, and glutamate neurotransmitter transporters in the solute-linked carrier (SLC) families SLC1 and SLC6, and other transporters in the SLC17, SLC18 and SLC22A families (*César-Razquin et al., 2015*). In addition, ATP-binding cassette transporters have been implicated in altered neurotransmitter distribution (*Borycz et al., 2008*). In *Drosophila*, these transporter families contain 137 members that we considered as candidates potentially involved in the histamine–carcinine cycle. To test their possible roles in photoreceptor neurons, we knocked down each candidate gene individually using Glass Multiple Response element (GMR)-Gal4 to drive double-stranded RNA expression in photoreceptors and evaluated visual signal transduction by ERG recordings.

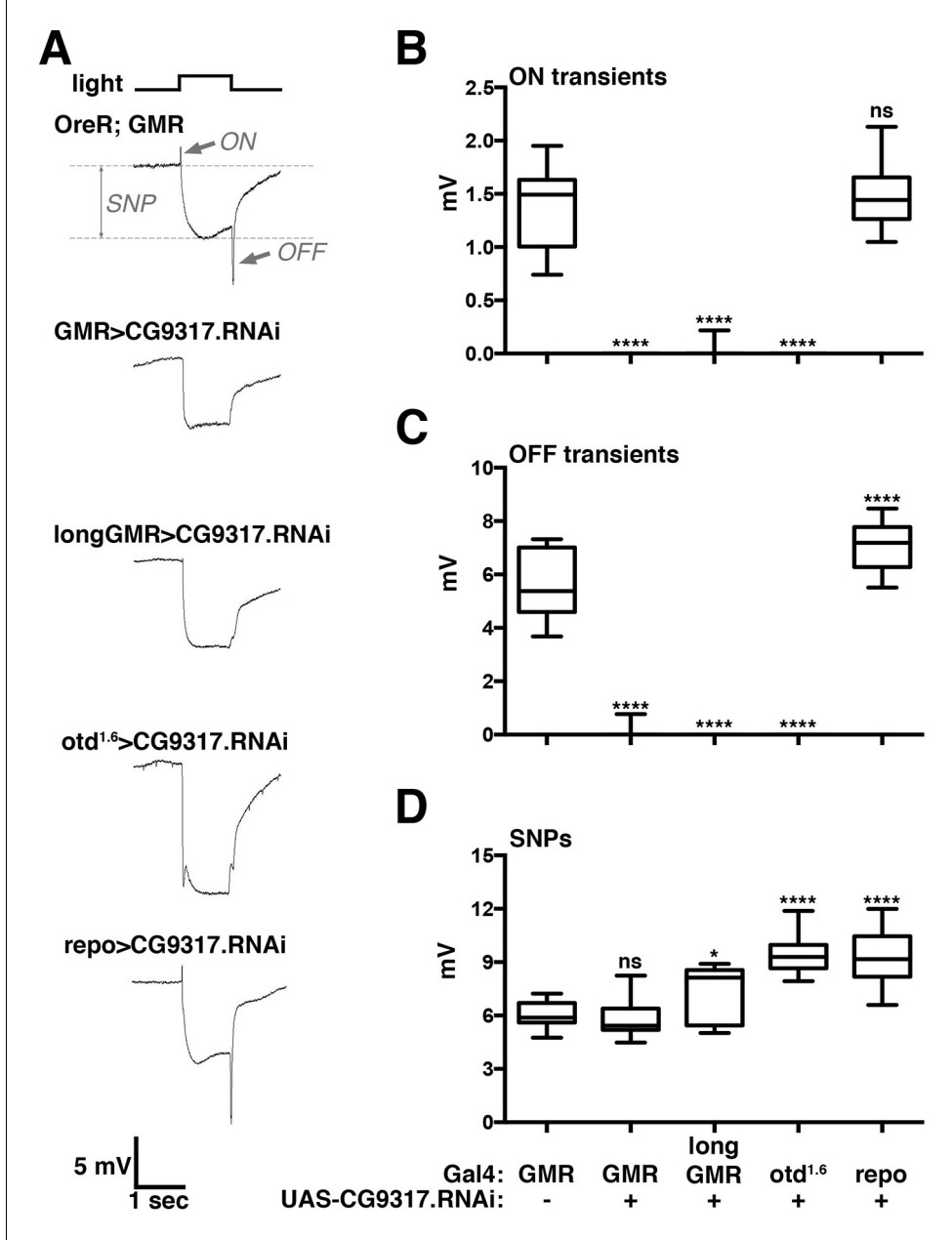

**Figure 1.** Photoreceptor specific knockdown of *CG9317* blocks visual neurotransduction. (**A**) ERGs recorded from female flies expressing a UAS-CG9317 RNAi transgene (VDRC 101145) targeting *CG9317* under the control of the indicated Gal4 driver specific for photoreceptors (GMR-Gal4, longGMR-gal4 and otd[1.6]-Gal4) or glia (repo-Gal4). Quantifications of (**B**) ON transients, (**C**) OFF transients, and (**D**) sustained negative photoreceptor potentials were averaged from three replicate experiments, including at least 45 traces from 15 flies. Arrows indicate SNP, ON and OFF transient of the control recording. Graphs report upper and lower quartiles (box) and minimum and maximum values (whiskers). ns, not significant; *$p < 0.05$, ****$p < 0.0001$ compared to *OreR;GMR-Gal4* control. ERGs, electroretinograms; SNPs, sustained negative potentials.

The following source data is available for figure 1:

**Source data 1.** List of transporter genes tested by GMR-Gal4 driven knockdown.

The screen yielded a single transporter gene, *CG9317*, whose knockdown caused severely reduced ON and OFF transients (**Figure 1A**). To test the consistency of this result in multiple genetic

backgrounds and confirm its specific requirement in photoreceptor neurons, we expressed the *CG9317* RNAi transgene with two additional photoreceptor-specific Gal4 lines, *longGMR-Gal4* (*Wernet et al., 2003*) and *otd$^{1.6}$-Gal4* (*McDonald et al., 2010*) and the pan-glial driver, *repo-Gal4* (*Sepp et al., 2001*). ERG recordings indicated that *CG9317* is necessary in photoreceptor neurons, but not glia, for proper histaminergic transduction in the visual system (*Figure 1A–C*). Of note, compared with wild type, none of these genotypes displayed significantly reduced sustained negative potentials ( *Figure 1D*), indicating that *CG9317* knockdown did not affect overall photoreceptor health (*Williamson et al., 2010*). Based on these results and the findings presented below, we will refer to *CG9317* as *carcinine transporter*, abbreviated *carT*.

### Photoreceptor-specific expression of CarT is sufficient for visual signal transduction

To test whether *carT* is the causal gene, the knockdown of which is responsible for loss of ERG ON and OFF transient components, we generated null alleles by clustered regularly-interspaced short palindromic repeats (CRISPR)/Cas9-mediated mutagenesis (*Gratz et al., 2013*; *Bassett et al., 2013*; *Yu et al., 2013*). We isolated three individual CRISPR-induced deletions: 11 bp for *carT*$^{16A}$, 26 bp for *carT*$^{16B}$ and 56 bp for *carT*$^{43}$ (*Figure 2A*). Each of these mutations resulted in a frameshift within the first transmembrane domain (*Figure 2B*) followed by a premature stop codon within 17, 19 and 1 bp, respectively. All three *carT* alleles were homozygous viable and fertile, but lacked the ON and OFF transients, indicating that *carT* is required for visual signal transduction (*Figure 2C,D*). For further analysis, we focused on the allele with the largest deletion, *carT*$^{43}$. To prove that loss of CarT function causes the ERG-phenotype in *carT*$^{43}$ mutants, we tested whether it could be rescued by cell type-specific expression of the CarT transcript. When driven in *carT*$^{43}$ mutant photoreceptor neurons by longGMR-Gal4, CarT restored ON and OFF transient components of ERG recordings (*Figure 2C,D*). By contrast, glial-specific expression of CarT failed to rescue ON and OFF transient defects (*Figure 2C,D*). To further probe CarT function, we generated a Myc-CarT transgene. This N-terminally tagged transporter was fully functional, as its expression in *carT*$^{43}$ also rescued defects in ON and OFF transients (*Figure 2C,D*). Taken together, these data indicate that expression of CarT in photoreceptor neurons is necessary and sufficient for visual signal transduction.

### Photoreceptor *CarT* is required for normal visual behavior

To examine whether the disruption of visual neurotransmission in *carT* mutants alters visual behavior, we used a phototaxis assay (*Benzer, 1967*). Wild-type flies were naturally phototactic (*Figure 2E*). However, flies with defects in the synthesis of histamine (*Hdc*$^{MB07212}$) or its recycling (*ebony*$^{1}$ or *tan*$^{1}$) were randomly distributed between the lit and dark arms of a T-maze, consistent with a lack of phototactic preference due to their inability to detect light (*Figure 2E*). Similarly, phototactic behavior was severely reduced in *carT*$^{43}$ flies (*Figure 2E*). Light preference of *carT*$^{43}$ flies was restored by expression of wild-type CarT in photoreceptor neurons under control of longGMR-Gal4 (*Figure 2F*). Together, these results indicate that CarT expression in photoreceptor neurons is necessary and sufficient for behavioral aspects of normal vision and electrophysiological properties of visual transduction.

### Carcinine accumulates in *CarT* mutant lamina

Interference with aspects of the histamine–carcinine cycle by different mutations causes distinct changes in the distribution of these two metabolites in the retina and lamina, although overall architecture of these tissues is not altered in *carT, ebony, tan* and *Hdc* mutants (*Figure 3A–D*) and references (*Chaturvedi et al., 2014*; *Borycz et al., 2008*). Consistent with their function in the histamine–carcinine cycle, characteristic increases in the levels of histamine in *ebony*$^{1}$ (*Figure 3C*) and carcinine in *tan*$^{1}$ heads (*Figure 3D*) were detected by antibody staining (*Chaturvedi et al., 2014*). When compared with wild type, *carT*$^{43}$ heads displayed no discernable alterations in histamine staining intensity or distribution (*Figure 3C*). However, carcinine levels were elevated in *carT*$^{43}$ lamina (*Figure 3D*, arrow) and medulla (arrowhead). Staining for carcinine in *carT*$^{43}$ lamina partially localized to the glia visualized by *repo-Gal4* driven *UAS-mCD8::RFP* (*Figure 3E*) suggested a build up of carcinine in the glia and the extracellular space in the lamina, consistent with a requirement of CarT in carcinine

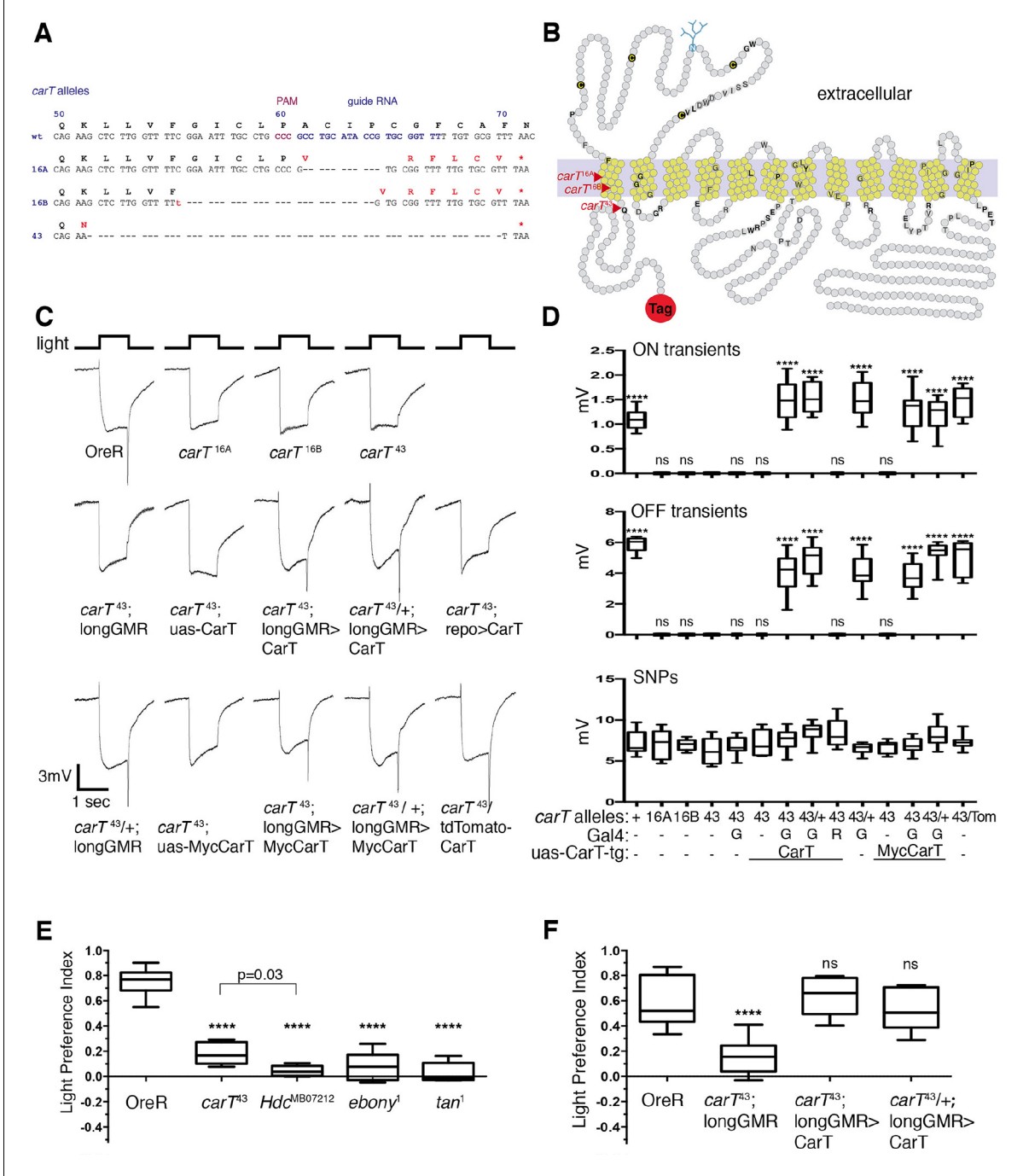

**Figure 2.** CarT is required in photoreceptors but not glia for normal cellular and behavioral visual responses. (**A**) CRISPR targeted site in the *carT* gene and resulting mutations. (**B**) Primary structure and predicted transmembrane domains (yellow) of CarT. Red arrows indicate location of premature stops within mutant alleles. Gray and black letters indicate conserved and highly conserved amino acids among SLC22 transporters (*Eraly et al., 2004*; *Koepsell, 2013*), respectively. The SLC22 family is characterized by a large extracellular loop containing four cysteines (C) and a glycosylation site (blue) at conserved positions. (**C**) ERGs recorded from female flies carrying wild type (+) or the indicated *CarT* alleles (16A, 16B, 43, or tdTomato-CarT) with or without expression of a wild-type or a Myc-tagged UAS-CarT transgene driven by *longGMR-Gal4* or *repo-Gal4* as indicated. (**D**) Quantifications of ON and OFF transients, and SNPs of all genotypes in panel C averaged from three replicate experiments, including at least 45 traces from 15 flies (G: *longGMR-Gal4*; R: *repo-Gal4*; -: not present). (**E**) Phototactic behavior of OreR and *carT*[43] mutant flies compared with other mutants that disrupt the histamine–carcinine cycle (*Hdc*[MB07212], *ebony*[1], or *tan*[1]) presented as a light preference index. (**F**) Phototactic behavior of flies expressing a UAS-CarT transgene in photoreceptor neurons under control of the *longGMR-Gal4* driver in *carT*[43] mutants compared with controls shows the restoration of wild type behavior. For each graph, the box outlines the upper and lower quartiles, and the whiskers show minimum and maximum recorded values. ns, not

*Figure 2 continued on next page*

*Figure 2 continued*

significant; ****p < 0.0001 compared with *carT*[43] (D) or OreR (E and F). CRISPR, clustered regularly-interspaced short palindromic repeats; ERG, electroretinograms; SNPs, sustained negative potentials.
The following figure supplement is available for figure 2:

**Figure supplement 1.** Phylogeny of the *Drosophila* SLC22 transporters.

transport into photoreceptors. Notably, carcinine accumulation in *carT*[43] brains was restored to wild-type levels by photoreceptor-specific expression of a Myc-CarT transgene (*Figure 3D*).

## CarT is enriched close to photoreceptor synapses

Within photoreceptors, the longGMR-driven Myc-CarT transporter was enriched in their axonal endings in the lamina (arrow in *Figure 3B*) and the distal retina (arrowhead in *Figure 3B*) indicating CarT may function near synaptic terminals and in neuronal cell bodies to facilitate perisynaptic and long-distance recycling, two pathways previously suggested (*Chaturvedi et al., 2014*; *Borycz et al., 2012*; *Rahman et al., 2012*). Endogenous CarT RNA expression is highly enriched in the eye (http://flybase.org/reports/FBlc0000157.html). To further examine endogenous CarT expression, we utilized CRISPR-induced deletions coupled with homology-directed repair to incorporate an in-frame insertion of tdTomato N-terminally to the CarT coding sequence (*Figure 2B*). ERG analysis of flies harboring one copy of this *tdTomato-CarT* over the nonfunctional *carT*[43] allele displayed normal ON and OFF transient responses, indicating that the endogenous tdTomato-tagged CarT was fully functional (*Figure 2C,D*). This endogenous tdTomato-CarT was highly enriched within photoreceptor projections in the lamina and medulla (*Figure 3F*). Co-labeling with the glial Ebony (*Figure 3G*), or synaptic Bruchpilot and a *3xPax3-eGFP*, which labels photoreceptor neurons (*Figure 3H*), indicates that CarT is present in photoreceptors near synaptic terminals (*Figure 3H*).

## CarT transports carcinine

To determine whether the CarT transporter utilizes carcinine as a substrate, we developed an immunofluorescence-based carcinine transport assay. Full-length or internally truncated Myc-CarT were expressed in S2 cells that were probed by antibody staining for carcinine uptake. When cultured in standard Schneider's medium, transfected S2 cells displayed no appreciable carcinine staining, indicating a lack of endogenous carcinine and sufficient specificity of the carcinine antibody in this context (*Figure 4A and A'*; quantified in D). When culture media was supplemented with carcinine (0.5 mM), cells expressing full-length Myc-CarT stained positively for carcinine, indicating its uptake in these cells. Importantly, neighboring untransfected cells (negative for the Myc epitope) lacked carcinine staining, despite being maintained in the same carcinine-supplemented medium as Myc-CarT positive cells (*Figure 4B and B'*). Furthermore, an internal deletion of 10 of the 12 transmembrane domains within CarT abolished carcinine uptake (*Figure 4C and C'*; quantified in D). Taken together, these data demonstrate CarT functions as a carcinine transporter.

Sequence comparison indicates that CarT is a member of the SLC22 family of transporters (*Eraly et al., 2004*) and together with other CarT-like transporters in invertebrates constitutes a distinct subfamily (*Figure 2—figure supplement 1*). Interestingly, OCT2 and OCT3, members of the closely related organic cation subfamily of SLC22A transporters, have been implicated in histamine uptake into human astrocytes (*Yoshikawa et al., 2013*) and the transport of histamine receptor antagonists (*Koepsell, 2013*). Many of the biochemically characterized members of the extended SLC22 family that are involved in the transport of neurotransmitters, exhibit overlapping substrate specificity providing transport redundancies (*Koepsell, 2013*). However, our data argue against such redundancy as *carT* null alleles efficiently blocks synaptic transmission dependent on the histamine–carcinine cycle (*Figure 4—figure supplement 1*). A role in this process has previously been suggested for the putative Na/Cl⁻ dependent neurotransmitter/osmolyte transporter Inebriated (*Stuart et al., 2007*), possibly by directly transporting carcinine into photoreceptors (*Gavin et al., 2007*). No direct evidence for such a function has been reported, however, suggesting that Inebriated may indirectly support the histamine–carcinine cycle by promoting the long-distance recycling of β-alanine from photoreceptors to glia (*Stuart et al., 2007*; *Chaturvedi et al., 2014*; *Borycz et al.,*

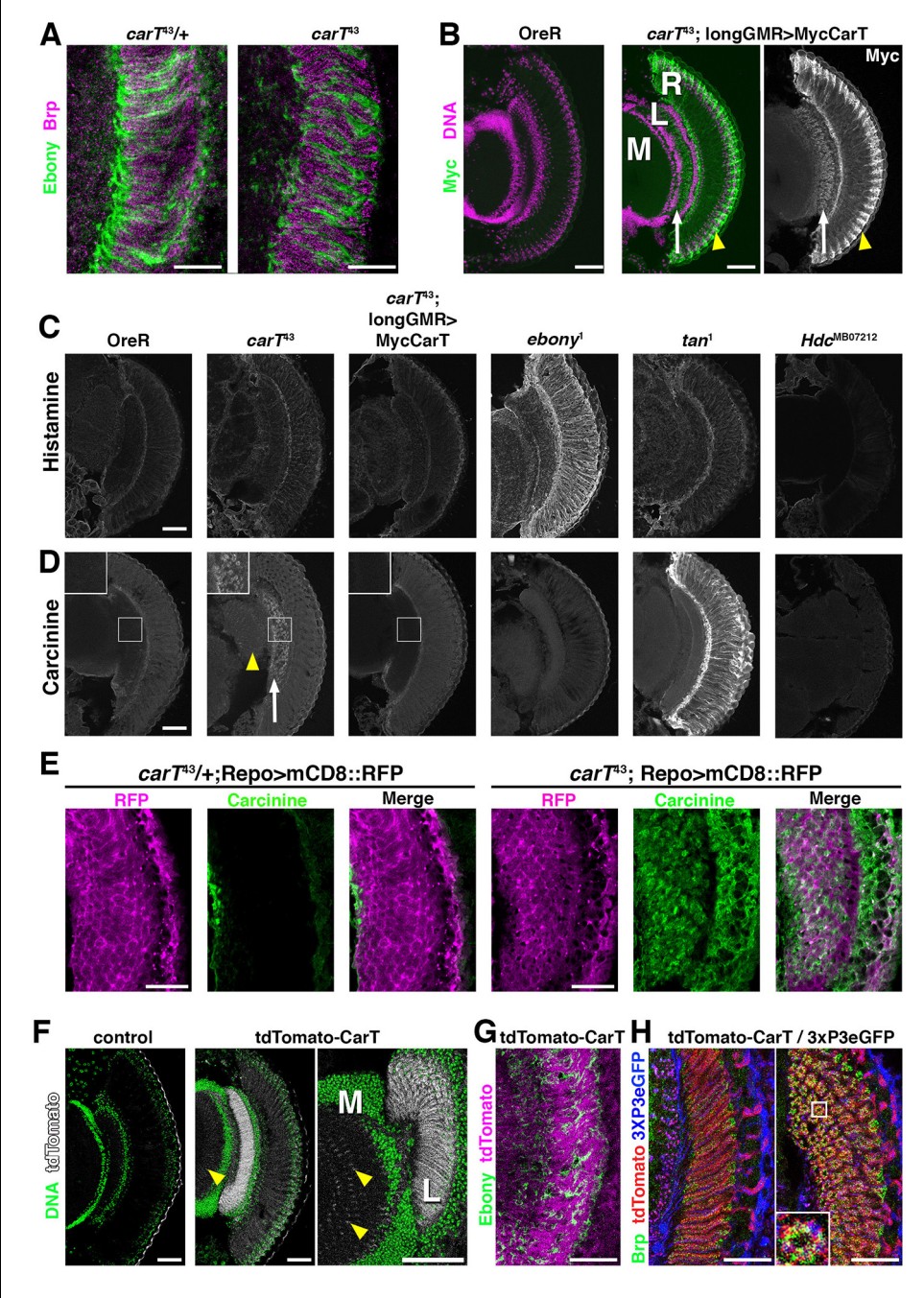

**Figure 3.** Loss of CarT in photoreceptors increases laminal carcinine. (**A**) Confocal sections of photoreceptor axonal endings within the lamina stained for the synapse marker Bruchpilot (Brp) and the glia-specific Ebony. (**B**) Micrographs of cryo-sections from control (OreR) and flies expressing the UAS-Myc-CarT transgene driven by longGMR-Gal4 in the *carT*[43] background. Retina (R), lamina (L), and medulla (M) neuropiles are stained for DNA (magenta) and Myc-tagged CarT (green). Arrow points to Myc staining in photoreceptor axonal endings and arrowhead to distal retina. (**C,D**) Micrographs of cryo-sections from control (OreR) and mutant flies affecting the histamine–carcinine cycle: *carT*[43], *carT*[43];GMR-Gal4/UAS-Myc-CarT, *ebony*[1], *tan*[1], and *Hdc*[MB07212] stained for histamine (**C**) or carcinine (**D**). Arrow and arrowhead in D point to carcinine accumulations in the lamina and medulla, respectively. (**E**) Micrographs from control (carT[43]/+ ) or *carT*[43] flies expressing a UAS-mCD8::RFP transgene driven by repo-Gal4 and stained for carcinine. (**F**) Micrographs showing tdTomato fluorescence of the in-frame tdTomato-CarT allele compared with wild type control. DNA staining is shown in green. Arrowheads point to tdTomato signal at R7 and R8 photoreceptor terminals within the medulla. (**G**) Confocal section of laminal

*Figure 3 continued on next page*

*Figure 3 continued*

region of a tdTomato-CarT fly stained for glia-specific Ebony. (H) Confocal sections of flies heterozygous for tdTomato-CarT (red) and photoreceptor-specific 3xPax3-eGFP (blue) stained for Brp (green). Sections are parallel to and across photoreceptor axons, respectively. Scale bars are 20 μm in A, E, G and H and 50 μm in B–D and F.

*2012*). An indirect role of Inebriated is also more consistent with its role in water homeostasis in the hindgut (*Luan et al., 2015*).

Here, we presented several lines of evidence that *CG9317*, which is highly expressed in heads, but not bodies (*Eraly et al., 2004*), encodes the carcinine transporter CarT. Heterologous expression of CarT in cultured cells was sufficient to facilitate carcinine uptake. This biochemical activity and the accumulation of carcinine in *CarT* lamina could be consistent with CarT facilitating export of carcinine from glia, or its import into photoreceptors. Strong support for the second possibility is provided by genetic experiments that reveal a requirement of CarT in photoreceptors, but not glia and the accumulation of carcinine in the lamina. CarT expression in photoreceptors is necessary and sufficient to sustain visual transduction, as indicated by ON and OFF transients as signatures of synaptic activity and by visual behavior. Together, these findings support a role of CarT as the transmembrane transporter responsible for the uptake of carcinine into photoreceptor neurons, a critical step in the histamine–carcinine cycle.

Histamine, similar to its impact on multiple behaviors in the mammalian brain (*Nall and Sehgal, 2014*; *Panula and Nuutinen, 2013*; *Shan et al., 2015*; *Castellan Baldan et al., 2014*), has been implicated in several circuits in the insect brain as well, including those controlling sleep, circadian rhythms and thermosensation (*Nall and Sehgal, 2014*; *Buchner et al., 1993*; *Oh et al., 2013*; *Hong et al., 2006*; *Suh and Jackson, 2007*). The identification of CarT's role in the histamine–carcinine cycle will provide additional tools to further address the role of histaminergic circuits in these different behaviors.

## Experimental procedures

### Fly work

Flies were maintained using standard conditions. Fly lines GMR-Gal4 (BL1104), longGMR-Gal4 (BL8121), repo-Gal4 (BL7415), *ebony*[1] (BL1658), *tan*[1](BL130), *Hdc*[MB07212](BL25260), UAS-mCD8::RFP (BL32218) were provided by the Bloomington *Drosophila* Stock Center. The *otd*[1.6]-*Gal4* line (*McDonald et al., 2010*) was a gift from T. Cook (Cincinnati Children's, Cincinnati, OH). The UAS-CG9317.RNAi line (VDRC KK101145) was obtained from the Vienna *Drosophila* Resource Center (VDRC). Other UAS-RNAi lines used for screening were obtained from either the VDRC or the Transgenic RNAi Project. CRISPR-mediated mutagenesis was performed via tools and protocols available from the O'Connor-Giles, Wildonger, and Harrison laboratories (www.flycrispr.wisc.edu). A germline mutation rate of 76.1% of injected flies was detected by polychromase chain reaction (PCR) amplification of the target region, slow ramp speed re-annealing, T7 endonuclease treatment, and polyacrylamide gel electrophoresis analysis of potential cleavage products (*Gratz et al., 2013*; *Bassett et al., 2013*; *Yu et al., 2013*). Exact lesions were characterized by sequencing of PCR products. Identified mutations were homozygous viable and crossed into a $w^+$ background prior to ERG recordings and behavioral assays. To generate *UAS-CarT* transgenic flies, a *CarT* complementary DNA was obtained form the Berkeley *Drosophila* Genome Project, cloned into a modified pUASt vector, sequence verified, and injected into embryos (Rainbow Transgenics, Camarillo, CA) using standard methods. Endogenous tdTomato was tagged by dual-cut CRISPR-mediated excision of 532 base pairs, including the CarT start site, and introduction of a donor plasmid containing an in-frame tdTomato sequence with roughly 1 kb homology arms (*Gratz et al., 2014*). Homology-directed repair resulting in the td-Tomato insertion was detected by PCR amplification of the tdTomato insert, and proper in-frame insertion was verified by sequencing.

## Electroretinogram recordings

ERGs were recorded as previously described (*Williamson et al., 2010*). In brief, voltage measurements of immobilized female flies were recorded with electrodes containing 2M NaCl placed on the corneal surface and inserted into the thorax. Measurements were filtered through an electrometer

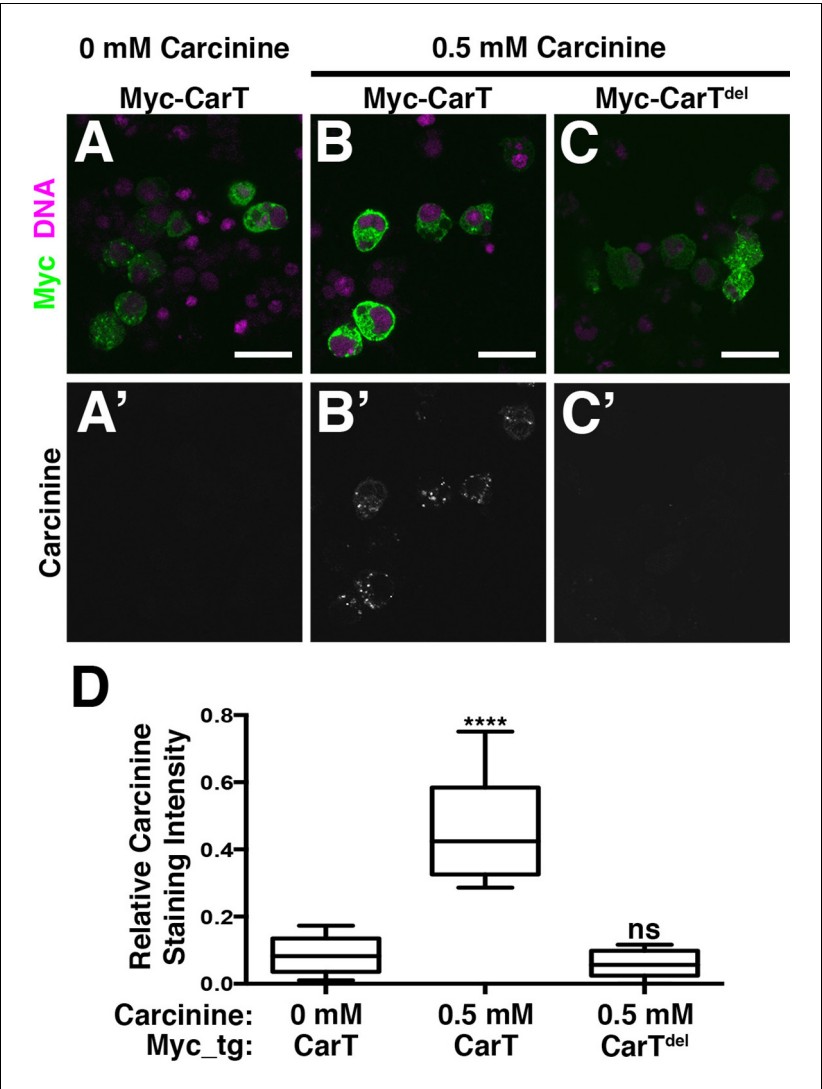

**Figure 4.** CarT is a carcinine transporter. (A) Micrographs of S2 cells transfected with the Myc-CarT transgene (A, A' and B, B') or the Myc-CarT^del transgene deleted for 10 internal transmembrane domains (C,C') cultured in media lacking carcinine (A,A') or supplemented with 0.5 mM carcinine (B,B' and C,C') and stained for DNA and the Myc epitope (A–C) or carcinine (A'–C'). (D) Quantification of carcinine signals normalized to signals for the Myc epitope. ****p < 0.001.

The following figure supplement is available for figure 4:

**Figure supplement 1.** Diagram summarizing the histamine–carcinine cycle and the proposed role of the CarT transporter in photoreceptor cells.

(IE-210; Warner Instruments, Hamden, CT), digitized with a Digidata 1440A and MiniDigi 1B system, and recorded using Clampex 10.2 and quantified with Clampfit software (Molecular Devices, Sunnyvale, CA). Light pulses (1 s) were computer controlled (MC1500; Schott, Mainz, Germany).

## Phototaxis assays

For light-choice assays (*Benzer, 1967*), male flies were collected within 12 hr of eclosion and aged for 3–4 d. Flies were anesthetized briefly with $CO_2$, transferred to empty culture tubes, and left to recover for 1 hr. For testing, a group of 20 flies was introduced into a T-maze apparatus and allowed to distribute for 20 s between a dark tube and a tube exposed to incandescent light. Each group

was tested in triplicate. A light preference index was calculated using the equation, *PI = (#Light-#Dark) / #total.* For all genotypes, at least 100 flies were tested.

## Histology

Fly heads were dissected in HL3 hemolymph-like solution (*Stewart et al., 1994*) to remove the proboscis and posterior cuticle, fixed for 4 hr in ice cold 4% 1-ethyl-3-(-3-dimethylaminopropyl) carbodiimide (wt/vol, Sigma, St Louis, MO) in 0.1 M phosphate buffer solution, washed overnight in 25% (wt/vol) sucrose in phosphate buffer (pH 7.4), embedded in optimal cutting temperature compound, frozen in dry ice, and sectioned at 20 µm thickness on a cryostat microtome (Hacker-Bright, Winnsboro, SC). Sections were incubated overnight with antibodies to histamine (1:1000, Sigma, St Louis, MO, cat# H7403 preabsorbed with 200 µM carcinine) or carcinine (1:1000, Immunostar, Hudson, WI, cat# 22939 preabsorbed with 200 µM histamine [*Chaturvedi et al., 2014*]). Other antibodies used include anti-Ebony (gift from Bernhard Hovemann [*Richardt et al., 2002*]), anti-Bruchpilot (nc82, Hybridoma Bank, Iowa City), anti-Myc (9E10, BAbCO, Richmond, CA), and anti-RFP (Rockland, Limerick, PA, cat# 600-401-379). Secondary antibodies were labeled with Alexa488 (1:500, Molecular Probes, Pittsburg, PA, cat# A-11008), Alexa568 (1:500, Molecular Probes, cat# A-11011), or Alexa647 (1:250, Molecular Probes, cat#A-21235). Where indicated, Topro-3 Iodide (Molecular Probes, T3605) was used to stain DNA. Images were captured with 20× NA 0.75 or 63× NA 1.4 lenses on an inverted confocal microscope (LSM510 Meta; Carl Zeiss, Oberkochen, Germany) at 21°C–23°C.

## Carcinine uptake assay

Because radioactive carcinine is not commercially available, we developed an immunofluorescence-based transport assay. *Drosophila* S2 cells were cultured using standard methods and transfected with plasmids containing Myc-tagged constructs as noted using the TransIT-2020 manufacturer's protocol (Mirus, Madison, WI). Transfected cells were incubated with 0.5 mM carcinine for 24 hr prior to a 1 hr fixation in ice cold 4% 1-ethyl-3-(-3-dimethylaminopropyl) carbodiimide (wt/vol, Sigma) in 0.1 M phosphate buffer solution. Cells were then permeabilized in phosphate-buffered saline containing 0.3% Saponin, blocked in 5% normal goat serum, and incubated overnight with carcinine and Myc antibodies. Quantification was performed in ImageJ by normalizing the integrated density of the carcinine signal by that of the Myc signal.

## Phylogenetics

CarT was identified as a SLC22 family member using BLASTpsi alignments against the NCBI non-redundant protein sequence database. Drosophila SLC22-related family members were identified from existing annotations on FlyBase and by running a BLASTp search with the CarT Isoform C peptide sequence against only *Drosophila melanogaster* (taxid:7227). Searches for individual organisms were performed using the following taxa identifiers on NCBI BLASTp suite: *Homo sapiens* (taxid:9606), *Mus musculus* (taxid:10090), *Apis mellifera* (taxid:7460). Additionally, previously identified mammalian SLC22 transporters were included (*Koepsell, 2013*; *Zhu et al., 2015*; *Martin and Krantz, 2014*). For preparation of multiple alignments, CLUSTAL Omega was used (http://www.ebi.ac.uk/Tools/msa/clustalo/) with default parameters (*Sievers et al., 2011*). Multiple sequence alignments were entered into Clustal W2 and phylogenies were generated using the Neighbor-joining clustering method with default parameters (*Larkin et al., 2007*). The resulting phylogenies were analyzed using Interactive Tree of Life (iToL; http://itol.embl.de), an online phylogeny tree viewer and editor (*Letunic and Bork, 2011*).

## Statistics

Statistical significance was determined using one-way analysis of variance (ANOVA) followed by Tukey's or Bonferroni's multiple comparisons using GraphPad Prism 6.

## Acknowledgements

We would like to thank Drs. Hong-Sheng Li and Tao Wang for communicating results before publication, and Drs. Aylin Rodan, Anna Ziegler and Bernhard Hovemann and the Krämer lab for helpful

comments and the Bloomington Stock Center for fly stocks, the Berkeley *Drosophila* Genome Project for producing DNA clones, the *Drosophila* Genomics Resource Center for their distribution. FlyBase provided important information used in this work. This work was supported by grants from the NIH (EY010199 and EY021922) to H. K., NIH training grant (5T32DA007290) to D.S., NSF Graduate Research Fellowship (1000176311) to A.T.M., and by core grant EY020799.

## Additional information

### Funding

| Funder | Grant reference number | Author |
|---|---|---|
| National Eye Institute | EY010199, EY021922 | Helmut Krämer |
| National Institute on Drug Abuse | DA007290 | Drew Stenesen |
| National Science Foundation | 1000176311 | Andrew T Moehlman |

The funders had no role in study design, data collection and interpretation, or the decision to submit the work for publication.

### Author contributions

DS, ATM, HK, Conception and design, Acquisition of data, Analysis and interpretation of data, Drafting or revising the article

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
