## [Decision Letter]

Thank you for submitting your work entitled "The carcinine transporter CarT is required in *Drosophila* photoreceptor neurons to sustain histamine recycling" for peer review at *eLife*. Your submission has been favorably evaluated by K VijayRaghavan (Senior editor), Hugo Bellen (Reviewing editor), and two reviewers, one of whom David Krantz, has agreed to reveal his identity.

The reviewers have discussed the reviews with one another and the Reviewing editor has drafted this decision to help you prepare a revised submission.

Summary:

The authors first performed an RNAi screen to knock down 132 transporter candidates and found that one – *CG9317* – which they rename *carT*, disrupts ERGs a standard measure of signaling in the visual system. They determined that expression was required in photoreceptors but not glia, then confirm their findings using CRISPR/CAS-9 mutants in *carT* and genetic rescue with a cart transgenes. They back up their electrophysiological results with both behavioral assays demonstrating phototactic deficits and carcinine build-up in *carT* mutants, both rescued with a *carT* transgene. Finally they prove that *carT* transports carcinine into S2 cells as predicted. The paper by Stenesen et al. very convincingly shows that the CarT transporter is required for carcinine uptake, which in turn is a reaction product of histamine modification in glia cells that take up histamine from the synaptic cleft. With this discovery, the circle for release and reuptake of this neurotransmitter has been closed for the first time.

Minor points that we would like to see addressed:

Minor concerns include some difficulty in visualizing the label for carcinine and histamine in Figure 3. It seems likely that this could result from the relatively low resolution of the PDF. Nonetheless, the authors should ensure that the images will be easily interpreted in the final version on the manuscript.

The authors should also comment on the overall distribution of endogenous *carT*, and whether or not this is known. The model in Figure 4 indicates that *carT* localizes to photoreceptors but not glia and this is clearly supported by their RNAi data. However, it seems possible that *carT* could be expressed in other neurons in the CNs and perhaps play a role in other organs.

Additional, small changes that might improve the paper for the interested reader include a list of the genes that were probed in their RNAi screen, and a brief note about the phylogeny of SLC22A family in the fly. The authors state: "[…] our data argue against such redundancy for CarT as it efficiently blocks the histamine-carcinine cycle." Nonetheless, it would be nice to know if there are any similar fly orthologs with the potential to compensate for loss of *carT*.

I think it would be reasonable to ask the authors to provide better colocalization data for CarT in the synaptic region of the lamina to support their core argument. Ideally one would like to know whether CarT resides at the capitate projections, invaginations of glia into the photoreceptor terminals, but an immuno-EM study of this is beyond the scope of the present work. However, colocalization analysis with glia markers, active zone markers and pre- and postsynaptic cell markers at high resolution would be helpful.

Similarly, it would be helpful to see carcinine accumulations in the mutant indeed occurring the glia cells, as predicted by the model, to support the core argument. Again, this would only require higher resolution and more detailed immunohistochemical analysis.

---

## [Author Response]

*Minor points that we like to see addressed:*

*Minor concerns include some difficulty in visualizing the label for carcinine and histamine in Figure 3. It seems likely that this could result from the relatively low resolution of the PDF. Nonetheless, the authors should ensure that the images will be easily interpreted in the final version on the manuscript.*

Indeed, the PDFs for the initial submission were too compressed; looking at the current images we are confident that the data presented in the final version will be easily interpreted. To aid the readers in interpreting the panels addressing carcinine accumulation in *carT* null flies (Figure 3 in revised version), we have included inserts with enlarged regions of similar positions from the lamina of the OreR, *carT^43^, and carT^43^;longGMR>MycCarT* genotypes. This will allow for a more focused comparison of this region, which displays increased carcinine staining in our *carT* mutant that is rescued by photoreceptor expression of the tagged carT transgene. Also, as outlined in our last comment below, we include an additional panel (Figure 3 in the revised version), which displays higher resolution images of the laminal carcinine increase in *carT^43^* compared to a heterozygous control.

The authors should also comment on the overall distribution of endogenous carT, and whether or not this is known. The model in Figure 4 indicates that carT localizes to photoreceptors but not glia and this is clearly supported by their RNAi data. However, it seems possible that carT could be expressed in other neurons in the CNs and perhaps play a role in other organs.

We now point out in the Results section that in adult flies *carT* RNA expression is highly enriched in the eye (subsection “CarT is enriched close to photoreceptor synapses”). Furthermore, to more accurately assess CarT distribution, we have now generated a tdTomato-CarT fusion in the endogenous genomic locus using CRISPR/Cas-9 mediated excision coupled with homology directed repair. In the current manuscript, we use ERG recordings to demonstrate that this allele is functional when heterozygous over the *carT^43^* null allele (Figure 2). Localization of this tdTomato-CarT fusion within the visual system is now displayed in Figure 3. In the course of these experiments we gained some initial indications that this transporter is also expressed in other tissues. We feel, however, that a careful description of the cell types expressing CarT in those tissues is beyond the scope of this “Short Report”. We will describe those expression patterns in a separate future manuscript.

*Additional, small changes that might improve the paper for the interested reader include a list of the genes that were probed in their RNAi screen, and*

The revised manuscript now includes a supplemental table to Figure 1, which lists the genes probed in the RNAi screen, their transporter family, and the stock designation of the specific RNAi line assayed.

*a brief note about the phylogeny of SLC22A family in the fly. The authors state: "[…] our data argue against such redundancy for CarT as it efficiently blocks the histamine-carcinine cycle." Nonetheless, it would be nice to know if there are any similar fly orthologs with the potential to compensate for loss of carT.*

We now include Figure 2—figure supplement 1, which depicts a phylogenetic comparison of SCL22A family members from the human, mouse, honeybee and *Drosophila* genomes. This analysis suggests the existence of a distinct invertebrate subclass of SLC22A family members with potential functions similar to CarT. However, as mentioned above, our CRISPR-induced mutants and the transgenic rescue experiments demonstrate *carT* is the causal gene affecting the identified phenotypes.

*I think it would be reasonable to ask the authors to provide better colocalization data for CarT in the synaptic region of the lamina to support their core argument. Ideally one would like to know whether CarT resides at the capitate projections, invaginations of glia into the photoreceptor terminals, but an immuno-EM study of this is beyond the scope of the present work. However, colocalization analysis with glia markers, active zone markers and pre- and postsynaptic cell markers at high resolution would be helpful.*

We agree with the reviewers and for that purpose have generated the endogenous tdTomato-tagged *carT* allele (Figure 3), which we used to perform co-labeling experiments with antibodies against the glial specific Ebony, the synaptic marker, Bruchpilot, and a photoreceptor specific 3xPax3-eGFP transgene (Figure 3). These data further support our argument that the major contribution of CarT to histamine recycling happens locally, close to photoreceptor synapses.

*Similarly, it would be helpful to see carcinine accumulations in the mutant indeed occurring the glia cells, as predicted by the model, to support the core argument. Again, this would only require higher resolution and more detailed immunohistochemical analysis.*

To address this issue, we performed transgenic labeling of glial cells by expressing a membrane bound uas-mCD8::RFP under Repo-Gal4 control in the *carT^43^* background and localized carcinine by immunohistochemistry (Figure 3). These data show that carcinine accumulates in and around glia consistent with a buildup of carcinine in response to the *carT* mutant-induced failure to transport carcinine into photoreceptors.